# Extreme Atlantic hurricane seasons made twice as likely by ocean warming

Peter Pfleiderer[a,b,*], Shruti Nath[a,c], and Carl-Friedrich Schleussner[a,b]

[a]Climate Analytics, Berlin, Germany
[b]Humboldt University, Geographie, IRI-Thesys, Berlin, Germany
[c]Eidgenössische Technische Hochschule Zürich, Zürich, Switzerland

**Correspondence:** Peter Pfleiderer (peter.pfleiderer@climateanalytics.org)

**Abstract.** Tropical cyclones are among the most damaging extreme weather events. An increase in Atlantic tropical cyclone activity has been observed, but attribution to global warming remains challenging due to large inter-annual variability and modelling challenges. Here we show that the increase in Atlantic tropical cyclone activity since the 1980s can be robustly ascribed to variations in atmospheric circulation as well as sea surface temperature (SST) increase. Based on a novel weather pattern based statistical model, we find that the forced warming trend in Atlantic SSTs over the 1982-2020 period has doubled the probability of extremely active tropical cyclone seasons. For the year 2020, our results suggest that such an exceptionally intense season might have been made twice as likely by ocean surface warming. In our statistical model, seasonal atmospheric circulation remains the dominant factor explaining the inter-annual variability and the occurrence of very active seasons. However, our study underscores the importance of rising SSTs that lead to more extreme outcomes in terms of cyclone intensity for the same seasonal atmospheric patterns. Our findings provide a new perspective on the contribution of ocean warming to the increase in recent hurricane activity and illustrate how anthropogenic climate change has contributed to a decisive increase in Atlantic tropical cyclone season activity over the observational period.

## 1 Introduction

Tropical cyclones (TCs) are highly destructive extreme weather events (MunichRe, 2021), with a notable increase in intensity and associated damages over recent decades (Kossin et al., 2013, 2020; Holland and Bruyère, 2014; Knutson et al., 2019). Under anthropogenically caused climate change the impact severity of TCs is exacerbated due to more extreme precipitation (van Oldenborgh et al., 2017; Reed et al., 2020) and increased risk of storm surges following from sea level rise (Lin et al., 2016), amongst others.

Whether the observed increase in TC intensity arises from a long-term trend related to global warming however remains unresolved. While climate models project an increase in TC intensities (Bhatia et al., 2018; Walsh et al., 2016; Knutson et al., 2020), a recent study suggests that after correcting for missing storm observations prior to satellite observation there is no robust long-term trend in Atlantic major hurricane counts (Vecchi et al., 2021).

TC formation and intensification mostly depends on the atmospheric environment which varies strongly on inter-annual and intra-seasonal time-scales. TC formation mainly requires low vertical wind shear and strong low-level relative vorticity (Frank

and Ritchie, 2001; Sharmila and Walsh, 2017) alongside some initial perturbation (Dieng et al., 2017). The maximal potential intensity of a storm mostly depends on the vertical temperature gradient from the ocean surface to the upper troposphere (Emanuel, 1987; Emanuel et al., 2013). Whether a storm reaches its maximum potential intensity is however strongly constrained by the large-scale atmospheric circulation. As a result, substantial uncertainties on the impacts of dynamical effects of global warming on changes in TCs globally still exist (Knutson et al., 2019, 2020).

Given the large uncertainty in forced atmospheric circulation changes, and assuming that these changes are small in comparison to internal variability (Trenberth et al., 2015) a promising way forward could be to focus on thermodynamically forced changes instead. Using a numerical TC forecast model, Reed et al. (2020) attributed a portion of the rainfall of hurricane Florence to thermodynamic effects of global warming. This study followed the story-line approach in which dynamical conditions of the weather event are reproduced for different counterfactual thermodynamic forcings. Such approaches are however restricted to individual events with clearly defined atmospheric conditions and cannot be directly generalized to seasonal TC activity (Reed et al., 2020).

For a more generalizable approach, the role of internal variability needs to be established and separated from the potential thermodynamic forcing (Shepherd, 2016). Climate models could be used to this extent (Sippel et al., 2019), but this would require a large ensemble of climate simulations with adequate TC representation. Alternatively, circulation analogues can be used. For example, Cattiaux et al. (2010) reproduced European winter temperatures based on observed circulation patterns and their influence on local temperatures.

Here we follow the idea of circulation analogues to construct a probabilistic tropical cyclone season emulator based on the empirically assessed influence of atmospheric circulation patterns over the tropical north Atlantic on TC activity. We find that the sequence of weather patterns throughout the main hurricane season (August-October) explains most of the inter-annual variability in number of storms and their intensities. The full observed variability in TC activity can be reproduced by including sea-surface temperatures (SSTs) over the main development region (MDR see fig. S6) as an amplifying factor for most intense TCs. Using counterfactual experiments, we furthermore investigate the extent to which trends in Atlantic SSTs contribute to highly active tropical cyclone seasons under current climatic conditions.

## 2 Data and Methods

### 2.1 Data & Preprocessing

For the classification of weather patterns we use mean sea level pressure (MSLP) and vertical wind shear (VWS) calculated as the difference between 200hPa and 850hPa eastward wind from the ERA5 reanalysis (Hersbach et al., 2020) over the period 1982-2020. Weather patterns are classified over the tropical north Atlantic (10W-90W and 10N-30N). For the following preprocessing we transform the data from the original 0.28°x0.28° to a 1°x1° grid. In order to remove the direct influence of TCs in the reanalysis data we replace the 3x3 grid-cell square area encompassing the center of the storm with the average of its surrounding 16 grid cells. Finally, we transform the data from a 1°x1° grid to a 2.5°x2.5° grid and average 6-hourly data to daily data.

For the construction and validation of the TC emulator, we use daily sea surface temperatures (SST) from the "Daily Optimum Interpolation Sea Surface Temperature" (DOISST) data set (Huang et al., 2021). SSTs are averaged over the Atlantic main development region (MDR) defined as 90W-20W and 10N-20N (see figure S6). Majority of Atlantic TCs originate and develop in this region. Since the MDR is commonly used in the literature, we choose to use it here even though it is slightly smaller than the region we use to classify weather patterns.

We use historical climate model simulations from the 6th phase of the "Coupled Model Intercomparison Project" (CMIP6) to estimate anthropogenically forced trends in Atlantic MDR SSTs over the period 1982-2014. A list of the used models can be found in table S1 in the supplementary information. As a reference for longer SST observations we use the "Hadley Centre Sea Ice and Sea Surface Temperature" data set (HadISST) is used (Rayner, 2003).

We use TC observations from the world meteorological organization (WMO) agency provided by the "International Best Track Archive for Climate Stewardship" (IBTrACS) database (Knapp et al., 2010, 2018). Only storms in the Atlantic basin that are classified as tropical storms are considered resulting in a total number of 454 storms. Following Bell et al. (2000), we use accumulated cyclone energy (ACE) as a measure of seasonal TC activity:

$$ACE = 10^{-4} \sum v_{max}^2 \qquad (1)$$

Where $v_{max}$ is the 6 hourly sustained wind speed in knots of storms that have at least tropical storm strength according to the Saffir-Simpson hurricane wind scale ($v_{max} > 34 \, kts$).

TCs are classified according to the Saffir-Simpson hurricane wind scale according to which TCs with sustained winds of more than 64 knots are named hurricanes and TCs with sustained winds above 96 knots are major hurricanes. Following the definitions of the "National Oceanic and Atmospheric Administration" (NOAA) national weather service (CPC, 2021), we classify Atlantic hurricane seasons into above normal seasons if they produce more than 126.1 ACE or extremely active seasons if the produced more than 159.6 ACE.

## 2.2 Daily tropical Atlantic weather patterns and sea surface temperatures

We use a self organizing map algorithm (SOM) to classify daily tropical Atlantic weather into 20 patterns. A SOM is an artificial neural network that is used for dimensionality reduction and can be applied to classify synoptic weather patterns (Hewitson and Crane, 2002). Here we reduce the highly dimensional information of mean sea level pressure (MSLP) and vertical wind shear (VWS) over a 2.5°x2.5° grid spanning 10N-30N and 90W-10W to a 5x4 map where each node represents a weather pattern (see figure S1-2). We use an initialization that is based on a principle component analysis to guarantee the reproducibility of the results.

To guarantee that both variables (MSLP and VWS) have equal weight in the classification we standardize the variables to the 1982-2011 mean and standard deviation. The combination of these two variables is a suitable choice for our application as TC formation and intensification strongly depends on VWS while MSLP is generally helpful to characterize the prevailing atmospheric circulation.

Some selected weather patterns are shown in figure 1: Strong TC activity is observed during weather pattern w0 which is characterized by a large low-pressure anomaly and nearly no vertical wind shear in the east of the MDR. Strong VWS in this region leads to fewer and weaker storms (see weather pattern w3). A strong high-pressure anomaly as in weather pattern w15 is similarly TC inhibiting. A weak pressure gradient from west to east with low VWS in the MDR is associated with high TC activity (w12). All 20 weather patterns are shown in the supplementary material (figures S1, S2, S3).

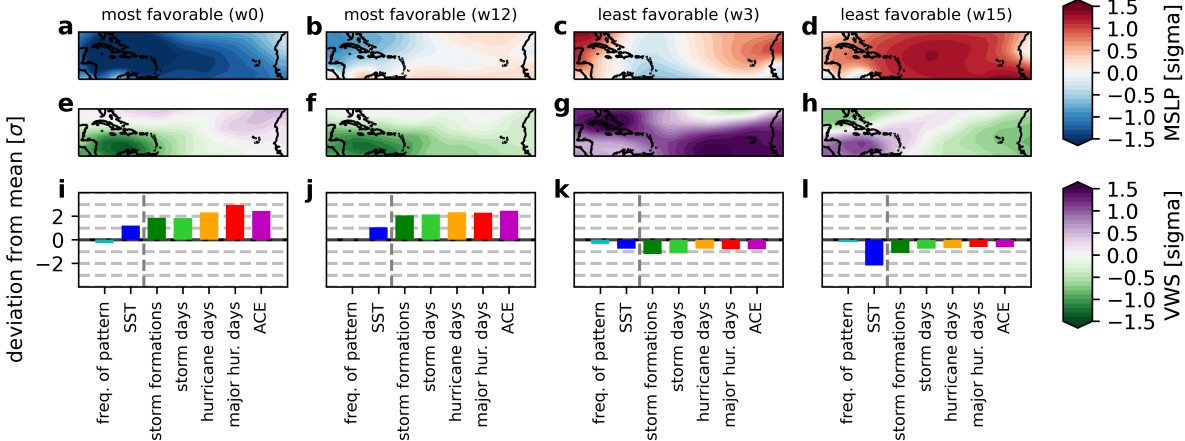

**Figure 1.** Tropical cyclone activity during selected weather patterns. Mean sea level pressure anomalies (**a-d**) and vertical wind shear anomalies (**e-h**) for four selected weather patterns w0, w12, w3 and w15. The last row (**i-l**) shows relative deviations from the average of all weather patterns expressed in standard deviations for the following statistics: frequency of the weather pattern, SST in the MDR, number of storm formations during the weather pattern, storm days, hurricane days, major hurricane days and average ACE generated during days with this weather pattern. A value of 2 indicates, that the statistic is 2 standard deviations higher during this weather pattern than for the average over all weather patterns. All 20 weather patterns are shown in figures S1, S2, S3

The intensities of TCs also depend on SSTs in the region. As shown in figure S7, strongest storms are found over warm SSTs. A quantile regression shows a significant relationship between warm SSTs and above median TC intensities. Weather patterns are not fully independent from SSTs: weather patterns with low pressure anomalies occur more often on days with warm SST anomalies (see figure 1 blue bars). However, no systematic association between SSTs and VWS is apparent (see figure S1 and S3). Although we will not be able to treat our weather patterns as independent from SST anomalies in the region, 100 both variables contain distinct information that is relevant for TC intensification.

### 2.3   Seasonal Tropical Cyclone Emulator

We construct a probabilistic emulator that creates series of storms with maximum sustained wind speeds for each day. TCs are rare events and their formation and intensification involves complex physical processes. In our emulator we break these processes down into three components that are fully independent from each other: i) storm formation, ii) storm duration 105 and iii) daily storm intensity. In these components the daily weather pattern slightly alters the probabilities for a new storm

formation and it's duration and the weather pattern in combination with regionally averaged SSTs alters the probabilities for intensification of an existing storm (see figure 2).

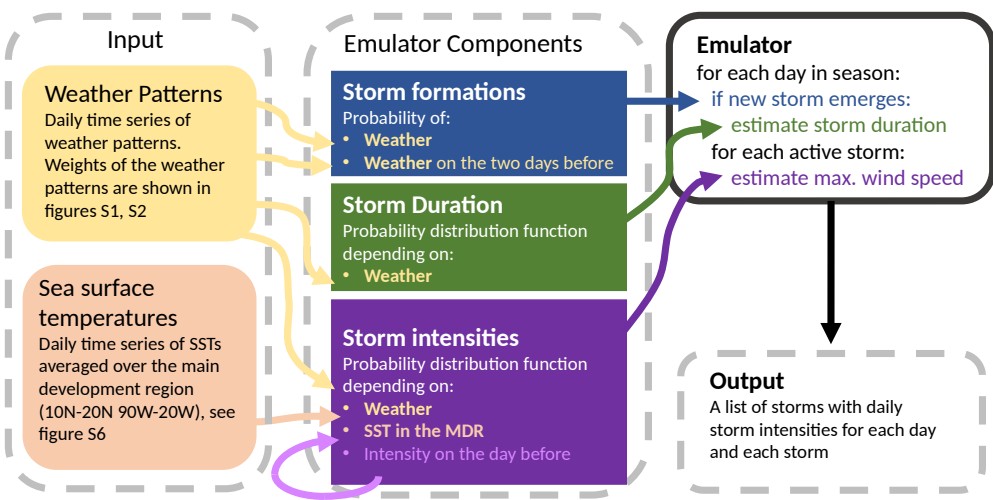

**Figure 2.** Schematic overview of the emulator. The input required to emulate TC seasons is shown on the left side. In the center, the three components of the emulator are listed. On the right side, the functioning of the emulator is sketched and the format of the output is indicated. Arrows between the left hand columns indicate which input is used in which components of the emulator. The light purple arrow indicates, that the estimation of storm intensities depends on the previous intensity of a storm.

### 2.3.1 Storm formation

The number of storm formations varies strongly between different large scale weather patterns (Jaye et al., 2019). Storm
formation predominantly occurs during weather patterns with low vertical wind shear, high relative humidity in the lower tro-
posphere and the existence of some kind of perturbation. The storm formation component relies on the following assumptions:
i) Weather patterns can favour or hamper storm formations (Jaye et al., 2019; Lee et al., 2018), ii) persistent weather conditions
can further increase or decrease formation probabilities.

Based on these assumptions we estimate the probability of a storm formation event $P_{gen}$ on a day $d$ with weather pattern
$w(d)$ as:

$$P_{gen}(d) = P_{obs}(gen|w(d)) \; * \; \frac{\sqrt{P_{obs}(gen_{+1d}|w(d-1)) \; * \; P_{obs}(gen_{+2d}|w(d-2))}}{P_{obs}(gen|all)} \tag{2}$$

The first factor $P_{obs}(gen|w(d))$ is the observed probability of storm formations for the given weather pattern $w$. The sec-
ond factor includes the probabilities of storm formations one and two days after the weather pattern that occurred one

$(P_{obs}(gen_{+1d}|w(d-1)))$ and two $(P_{obs}(gen_{+2d}|w(d-2)))$ days earlier respectively. These probabilities are given less weight by applying a square root and the factor is normalized by a dividing by the overall observed storm formation probability $(P_{obs}(gen|all))$.

### 2.3.2 Storm duration

There are numerous processes that can weaken and eventually dissipate TCs. The most common end of a TC is landfall. As we do not have information about the location of storms in our emulator, estimating the duration of a storm is challenging. For the development of this component we use the following assumptions: i) storms dissipate when making landfall, ii) the time a storm has before making landfall is modulated by its formation location and iii) the formation location is to some extent influenced by weather patterns (see figure S4 and S5).

We incorporate this dependence of storm duration on weather patterns on the day of storm formation by sampling the duration $D$ of a storm $s$ from a Gaussian kernel estimate $f_g$ of all storms that have formed during the weather pattern during which the storm has formed $w(d_f)$ and all neighboring weather patterns.

$$D(s) = f_g(D_{obs}[w_{row} - w_{row}(d_f) < 2 \ \& \ w_{col} - w_{col}(d_f) < 2]) \tag{3}$$

### 2.3.3 Storm intensity

We quantify storm intensity through the daily maximum sustained wind speed. We use the following assumptions for our daily storm intensity emulations: i) intensification can be favored or hampered by specific atmospheric circulation patterns (Frank and Ritchie, 2001; Lee et al., 2016), ii) the intensity of a storm depends on the intensity on the day before, iii) warmer SSTs in the MDR favour the intensification of intense TCs (Bhatia et al., 2018; Trepanier, 2020), iv) the relationship between SSTs and storm strength can be regularized by a quantile regression (see figure S7).

Assessing probability density functions for daily storm intensities for all possible combinations of weather patterns, SSTs, and storm intensities on the day before is challenging given the insufficient number of storm observations. Therefore, instead of estimating a PDF for the daily intensity from all observations that match to certain conditions (e.g. weather pattern w6, 28°C SST and 60kts wind speed on the day before) we estimate the intensity PDF from the 100 storm observations that are most similar to these conditions.

Furthermore, the distribution of observed intensities is skewed towards weak storms which would result in a low intensity bias in a straight forward application of the nearest neighbors approach (see figure S10a-c). We therefore introduce a linear relationship between regionally averaged SSTs and storm intensities (see quantile regression in figure S7) to guarantee that the intensities from which we sample are not systematically too weak.

For a given $SST_{target}$ we transform all observed storm intensities to artificial pseudo-intensities $v_{shifted}$ using the slope $\beta_\tau$ of the next quantile $\tau$ below the observed storm strength $v(s,d)$:

$$v_{shifted}(s,d,SST_{target}) = v(s,d) + \beta_{\tau(s,d)} * (SST_{target} - SST_{obs}(d))$$

$$\tau(s,d) = min(\tau : v(s,d) > \beta_\tau + c_\tau) \tag{4}$$

We use the Euclidean distance metric applied on standardized variables to identify the 100 nearest neighbors in terms of weather pattern and storm intensity on the day before:

$$D(d_i, d_j)^2 = \frac{(w(d_i) - w(d_j))^2}{\sqrt{\frac{1}{N}\sum_m (w(d_m) - \overline{w})^2}} + \frac{(v_{shifted}(d_i-1) - v_{shifted}(d_j-1))^2}{\sqrt{\frac{1}{N}\sum_m (v_{shifted}(d_m) - \overline{v_{shifted}})}} \tag{5}$$

For the weather patterns, which are not a continuous variable, we consider their coordinates in the SOM grid as locations and calculate differences between weather patterns as the sum of the squared differences in row and column numbers.

$$w(d_i) - w(d_j) = \sqrt{(w_{row}(d_i) - w_{row}(d_j))^2 + (w_{col}(d_i) - w_{col}(d_j))^2} \tag{6}$$

## 3 Results

### 3.1 Validation of the emulator

Figure 3 sketches the functioning of the emulator for three Atlantic hurricane seasons: 2020 was a highly active season with predominantly warm SSTs and favorable large-scale weather conditions allowing for strong TCs throughout most of the season.

2009 had similarly warm SSTs but less favorable weather conditions resulting in overall fewer days with strong storms. 1983 was an El Niño year with cool SSTs in the tropical Atlantic and mostly unfavorable weather conditions for TCs in the Atlantic basin. The chance of finding storms and especially the chance of finding major hurricanes in simulations (fig. 3d) for the respective years reflects the observed weather patterns and SSTs.

To validate the emulator, we re-simulated every hurricane season between 1982 and 2020 1000 times using the observed

sequence of daily weather patterns and SST averages over the MDR. We construct a new emulator for each decade using all the years but the decade we want to re-simulate as training data.

Large scale weather patterns are sufficient to explain most of the inter-annual variations in the number of storm formations (see fig. 4A). The remaining spread between individual simulation runs is to be expected as tropical storm formations have a strong stochastic component. Besides favorable weather conditions, storm formation requires a (small scale) perturbation in the

170 atmospheric flow such as African easterly waves to be initiated (Dieng et al., 2017), information that is lacking in our emulator.

The number of storm days per season is strongly related to the number of storms. The simulated storm durations enhance the representation of number of storm days resulting in a accurate representation of storm durations (see fig. S5) and a Pearson correlation coefficient of 0.69 between observations and the median of all simulations (see fig. 4B).

Finally, the storm intensity component produces a variety of storm intensities including major hurricanes (see fig. 4C). As

for the number of storm days the number of strong storms is tightly linked to the number of storm formations. But as storm

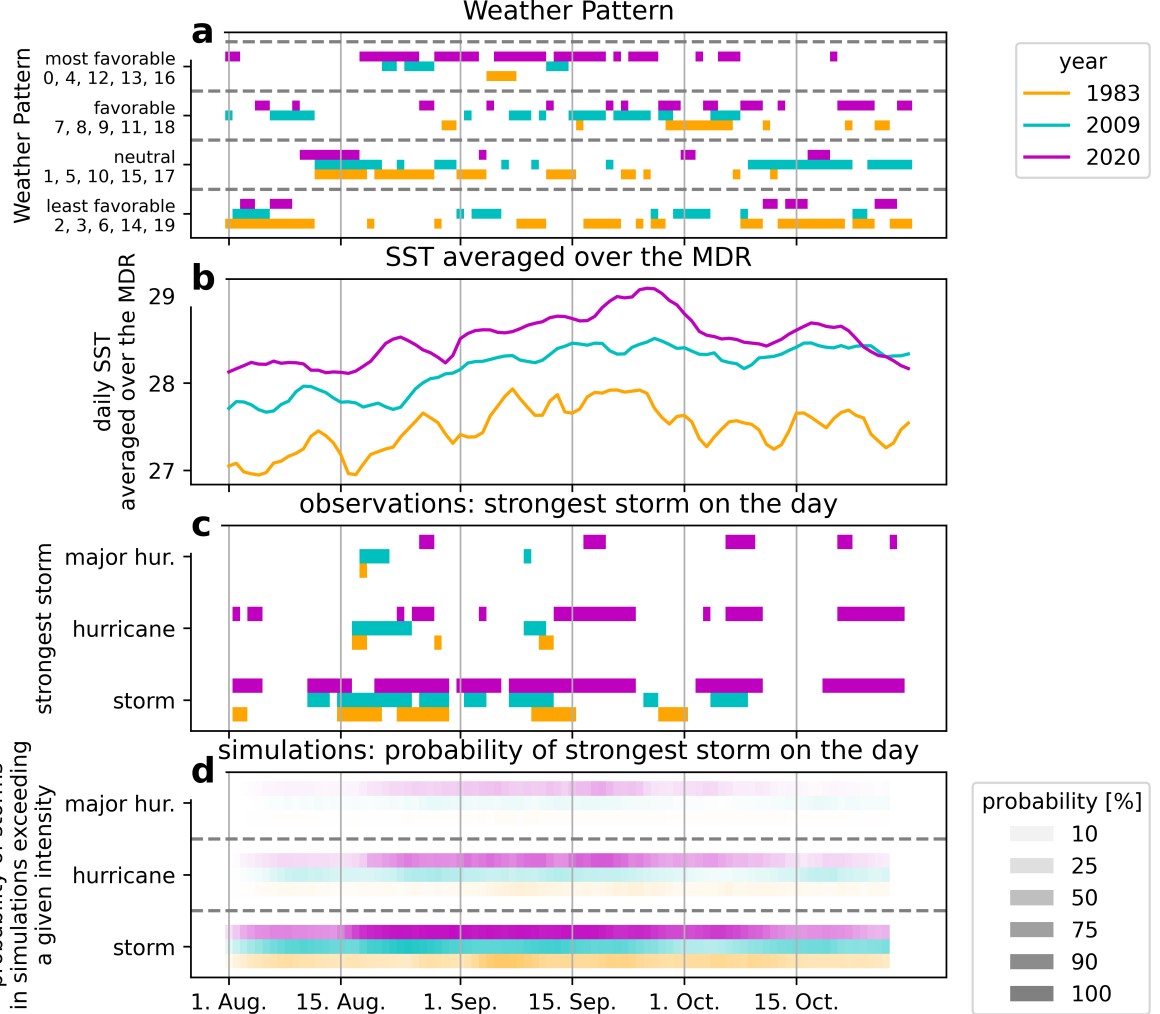

**Figure 3.** Functioning of the tropical cyclone emulator. **a**, Sequence of daily weather patterns grouped into four categories from least favorable for TC formation and intensification to most favorable for the years 2020 (purple), 2009 (cyan) and 1983 (orange). **b**, Daily SSTs averaged over the main development region for the same years. **c**, Intensity of the strongest storm for each day grouped into the categories storm, hurricane and major hurricane. **d**, Probability of exceeding the intensity thresholds of c in simulations from the emulator.

intensification is favored by certain weather patterns and warm SSTs, the potential for intensification alters between years. In combination, this results in an adequate representation of inter-annual variability in seasonal accumulated cyclone energy (ACE) as shown in figure 4D.

According to the correlation coefficients, major hurricane counts are slightly better represented than storm counts. While the number of major hurricanes is tightly linked to the amount of storm formations, the storm intensity component is an additional

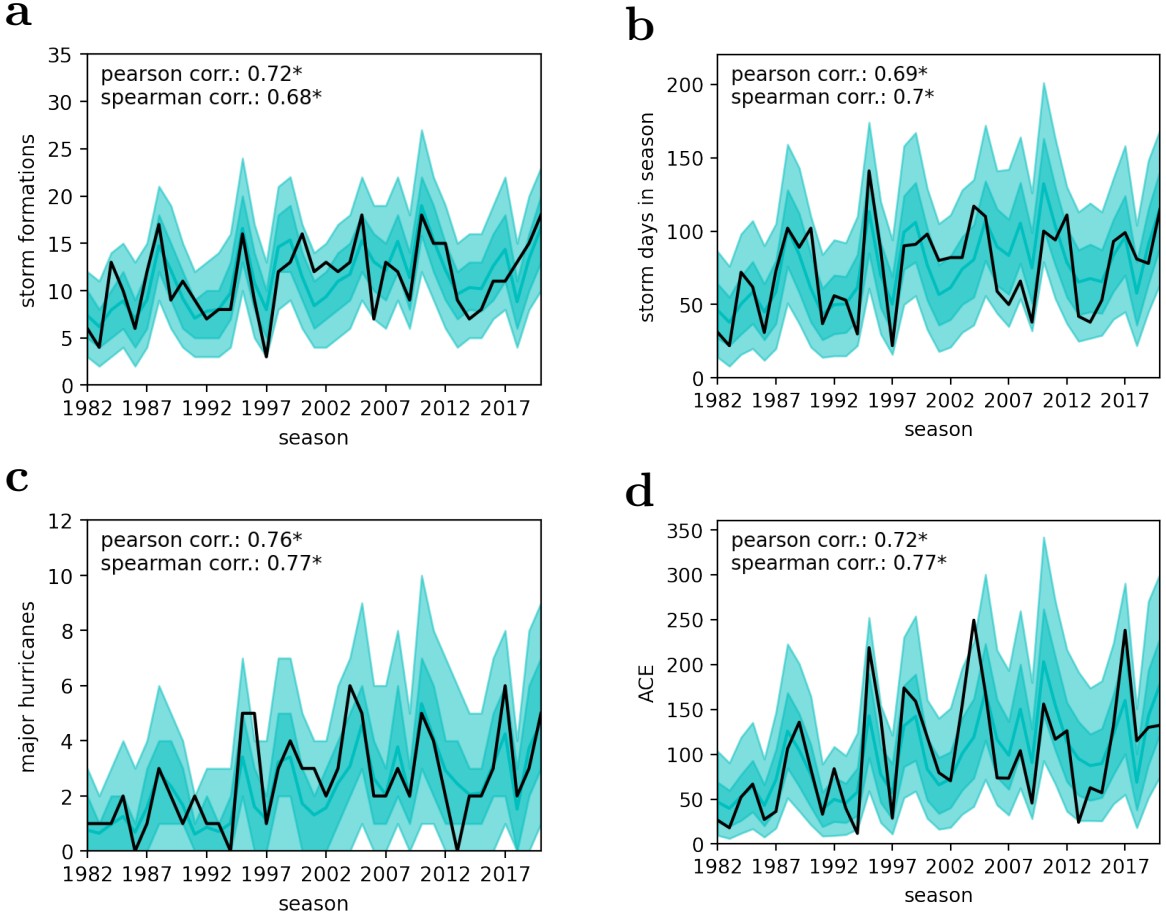

**Figure 4.** Cross validated hurricane season emulations. **a**, Number of storms as observed (black) and simulated (cyan). The light shading shows the 95% range of the 1000 simulations, the darker shading shows the 66% range. The mean is indicated by a solid line. The cross validated Pearson (Spearman) correlation coefficient between hindcasts and observations is indicated in the legend (see methods for more details on the decadal cross validation). **b**, As **a** but for the number of storm days in a season. **c**, As **a** but for the number of major hurricanes in a season. **d**, As **a** but for the seasonal ACE.

instance controlling under which conditions many major hurricanes are likely. The higher correlation for major hurricane counts is therefore an indication for a meaningful treatment of storm intensification in the emulator.

## 3.2 Sensitivity analysis

Strong simplifications were required to emulate TCs based on sequences of weather patterns and regionally averaged SSTs. The assumptions on which our methodological choices are made are plausible and appear to work well, they are however not without alternatives. We therefore test how alternative emulators perform.

The most critical part is the treatment of SSTs in the intensification component of the emulator as it directly influences some of the results. As shown in figures S11, emulators without any SST influence have considerable trends in residuals for major hurricane counts of 0.3 per decade (or 0.9 per Kelvin of seasonal SST, see fig. S11d). This misrepresentation in major hurricanes translates into a negative trend in seasonal ACE residuals (see figure S12). Including SSTs into the intensification component of the emulator reduces these trends significantly which suggests, that SSTs contain information that is required for an adequate representation of strongest TCs.

A simpler way of including SSTs in our emulator could be to estimate intensity probability functions directly from the 100 nearest neighbors in terms of weather patterns, storm intensities on the day before and SSTs. While this approach works well in the range of average conditions, there are systematic deviations between the nearest neighbors and the target conditions for more extreme conditions. As shown in figure S10d-e there is a warm bias for cool SSTs and vice versa which is a result of to few observations from which the nearest neighbors can be searched. Similarly, there is a bias towards weaker storms in the nearest neighbors (see figure S10a-c). Reducing the number of nearest neighbors from 100 to 20 only slightly reduces these biases. Ultimately, this results in a lack of sensitivity in our emulator.

In the supplementary information we present a number of additional emulators with slightly altered storm formation (see figure S8), storm duration (see figure S9) and storm intensity components (figures S10-S14). Most of these altered emulators yield similar results which supports the robustness of our results.

## 3.3 The effect of ocean warming on recent TC activity

We deploy the emulator to assess the contributions of large scale atmospheric circulation and forced warming of tropical Atlantic SSTs towards the likelihood of extremely active hurricane seasons. According to DOISST and over the period 1982-2020, SSTs in the Atlantic MDR have warmed at a rate of 0.3 $K$ per decade (figure 5b). This trend is slightly weaker in the HadISST dataset (figure 5a) for which also the global trend in SSTs over the period 1982-2020 is weaker than in other SST datasets (Yang et al., 2021). Using CMIP6 historical simulations we estimate that the forced trend on SSTs in the MDR throughout the hurricane season is 0.22 $K$ per decade for the period 1982-2014 (fig. 5a). Thus, the observed SST trend over the 1982-2020 period is to a large extent forced by global warming.

To disentangle forced changes in TC activity from internal variability we construct counterfactual scenarios in which we first remove the forced SST trend as estimated from CMIP6 simulations for the period 1982-2014. We then shift these detrended SST time-series so that on average they match the values of the forced trend for the years 2020 and 1982 and call these

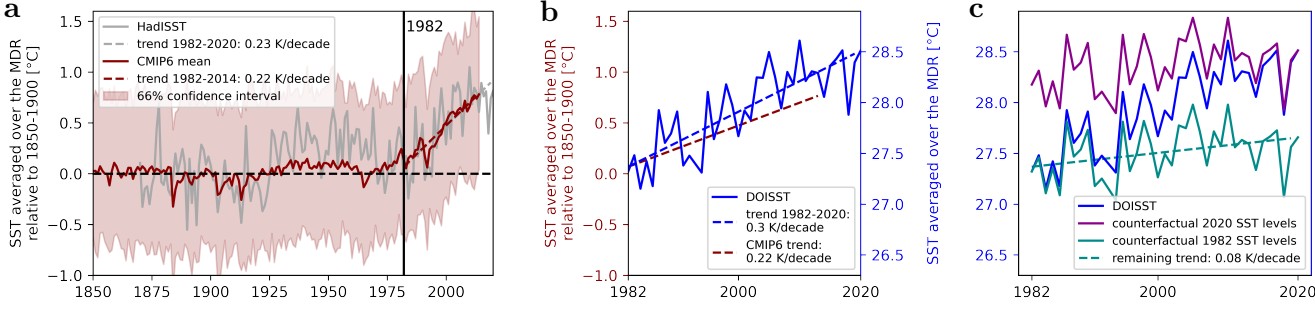

**Figure 5.** Sea surface temperatures averaged over August-October and the MDR. **a**, Ensemble mean of historical CMIP6 simulations (red) relative to 1850-1900 and HadISST observations (gray) relative to 1870-1900. The 66% range of the CMIP6 ensemble is represented by the red shading. Linear trends for CMIP6 (HadISST) over the period 1982-2014 (1982-2020) are indicated by dashed lines. **b**, DOISST observations for the period 1982-2020 in blue and respective to the right y-axis. Besides the linear trend in DOISST the linear trend of CMIP6 is indicated by a red dashed line using the left y-axis. **c**, Counterfactual SST scenarios based DOISST observations from which the CMIP6 trend is removed. This detrended SST time series is shifted to the value of the CMIP6 trend in the year 1982 (cyan) and the year 2020 (purple). The remaining linear trend in these counterfactual scenarios is indicated by a dashed line.

artificial SST time series *2020 scenario* and *1982 scenario* (figure 5c). The counterfactual SST scenarios contain the observed
215 year to year variations that can be linked to natural modes of variability such as ENSO. The only difference between these two
scenarios is that the 2020 scenario has 0.9 $K$ warmer SSTs than the 1982 scenario.

Both counterfactual scenarios contain a small linear trend of 0.08 $K$ per decade (fig. 5c). This remaining trend reflects that
the observed trend in DOISST is not solely due to global warming but that natural climate variability also contributes to the
trend over the period 1982-2020.

In the CMIP6 historic simulations, no forced warming in MDR SSTs is simulated for the period before 1980. The long
term average over the period 1850-1900 (27.27 $K$) is close to the value of the 1982-2014 trend in the year 1982 (27.32 $K$).
Therefore, SSTs in the counterfactual 1982 scenario are similar to pre-industrial levels for the MDR. Despite differences in the
1982-2020 trend, the HadISST dataset confirms the findings that SSTs in the MDR have not warmed considerably before the
1980's.

Over the period 1982-2020, large scale atmospheric circulation patterns are the dominant factor explaining year to year
variability in TC activity. Our emulations show high TC activity in the same years irrespective of the counterfactual SSTs (see
fig. 6b). For instance, the low activity in the years 1982-1987 is also simulated in the 2020 scenario while the years 1995, 2005,
2010 and 2017 have a high likelihood of becoming an extremely active season also in the 1982 scenario.

Nevertheless, differences in seasonal TC activity are apparent between the two scenarios. On average, the seasonal activity
is 25 ACE lower in the 1982 scenario as compared to the 2020 scenario (see fig. 6c). As a result, more than one third of the
seasons that are simulated to be above normal seasons in the 2020 scenario are below normal seasons in the 1982 scenario
(an above normal seasonal activity being defined as > 126.1 ACE (CPC, 2021)). Similarly, the number of simulations that are

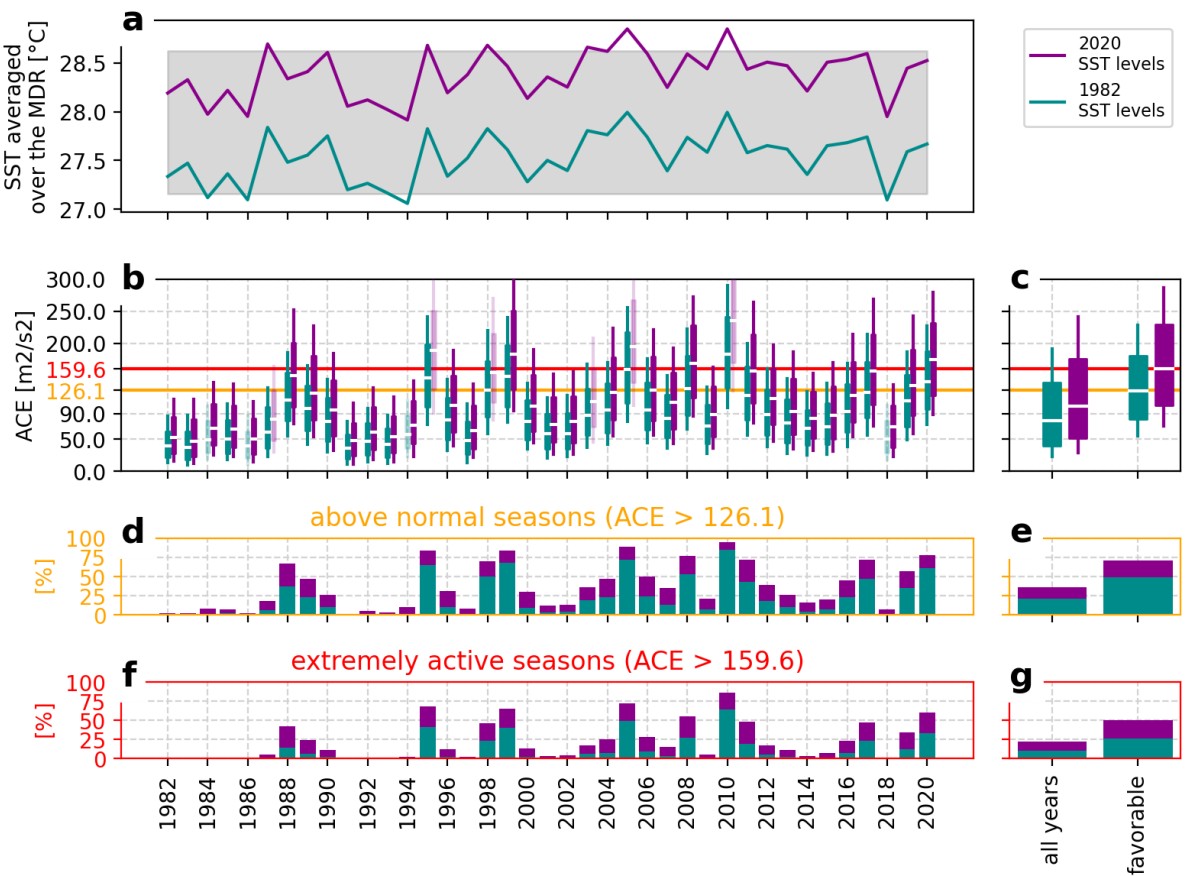

**Figure 6.** Atlantic hurricane seasons under different counterfactual SST scenarios. **a**, Two counterfactual SST scenarios: SSTs from which the forced SST trend has been removed and that are shifted to 2020 SST levels (purple) and shifted to 1982 SST levels (cyan). The gray recantangle indicates the range of observed seasonal SST averages. See figure 5 for more details. **b**, Simulations for the counterfactual scenarios of **a** displayed as boxplots. For years, where the seasonal SST averages in the counterfactual scenario are outside of the range of observed seasonal SST averages, the simualtions are shown in lighter shading. **c**, Simulations for all years aggregated and for the most favorable years defined as years for which half of the simulations in the 2020 SST scenario have more than 126.1 ACE. **d**, Probability of above normal seasons (ACE > 126.1). **e**, As **d** but for all years and favorable years. **f**, as **d** but for extremely active seasons (ACE > 159.6). **g**, As **e** but for extremely active seasons.

classified as extremely active (with ACE > 159.6) doubles from 11% in the 1982 scenario to 22% in the 2020 scenario (see fig. 6g).

Differences between the counterfactual scenarios are stronger in years with high TC activity (see fig. 6c). For years in which half of the simulations of the 2020 scenario are above normal seasons, the simulations are on average 36 ACE more active in the 2020 scenario than in the 1982 scenario. For these years the risk of finding an extremely active season (with ACE > 159.6) drops from 50% under current climate to 27% in the 1982 scenario (see fig. 6g). Our results do not imply that increasing sea-surface temperatures lead to more TC formations – but point towards a trend of more extreme outcomes for seasons with

many TCs. This is in line with a global trend towards more intense tropical cyclones over the observational record as well as projections (Masson-Delmotte et al., In Press).

Simulations from the emulator can moreover be used to analyze contributions to extremely damaging hurricane seasons such as that of 2020. 2020 is one of the most active recorded hurricane seasons with an ACE index of 178 and 5 major hurricanes in August-October (and two additional major hurricanes in November). The season was characterized by weather patterns that

are favorable for TC formation and intensification and relatively warm SSTs. Figure 7a shows the probabilities of finding such a season under counterfactual SST scenarios.

Compared to other years, 2020 has a high probability of becoming an above normal season (78%), a considerable probability of becoming an extremely active season (60%) and 47% of the simulations reach the observed ACE of 178 (see fig. 7a).

Under a counterfactual 1982's SST scenario with similar modes of internal climate variability, weather patterns and short

term variations in Atlantic SSTs, the season would have less likelihood to become an above normal hurricane season (61%), an extremely active season (33%) or even a season with 178 ACE (21%).

For 2020 weather conditions, the warming of Atlantic SSTs since the 1980's has increased the probability of finding a season with 178 ACE by a factor of 2.2 (see fig. 7b). For a year like 2005 which according to our analysis had a higher likelihood of becoming an extremely active season than 2020, the probability of finding a seasonal ACE of 178 is a factor of 1.6 higher in

the 2020 scenario as compared to the 1982 scenario. The likelihood of finding 178 ACE in any year irrespective of the weather conditions is increased by a factor of 2.4. The increase in likelihood of finding 178 ACE is higher for seasons with weather conditions that are hampering TC formation and development. For a year like 1983 with very few TC formations there are no simulations that reach 178 ACE in neither of the counterfactual scenarios.

## 4  Discussion and conclusions

We have demonstrated that the observed Atlantic tropical cyclone activity over the last 40 years can be reproduced with a probabilistic emulator based on large-scale weather patterns and SSTs. Over this period, we observe a trend in weather patterns favoring more active TC seasons. Whether or not this trend in atmospheric circulation can be attributed to anthropogenic climate change or other external drivers such as aerosol loading's (Dunstone et al., 2013) remains an open question.

It is important to highlight that our weather patterns and regional SST time series are not fully independent. Specifically, it

appears that years with warm Atlantic SSTs are also years where Atlantic SSTs are warm relative to the rest of the tropics and it

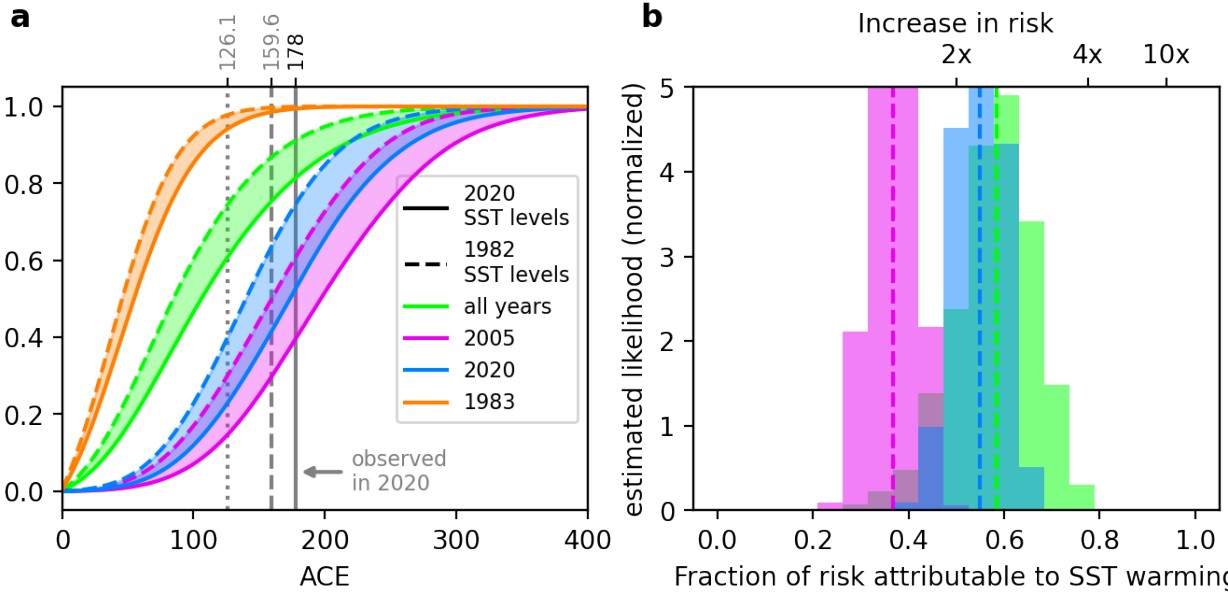

**Figure 7.** Influence of ocean warming on the hurricane season 2020. **a**, Cumulative distribution functions for seasonally aggregated ACE for the 2020 scenario (solid) and the 1982 scenario (dashed). All years between 1982-2020 aggregated in green, 2005 in purple, 2020 in blue, and 1983 in orange. The area between the 2020 scenario and the 1982 scenario is shaded. The horizontal gray lines indicate 178 ACE which was observed in 2020 (solid), the threshold for above normal seasons 126.1 (dotted) and the threshold for extremely active seasons 159.6 (dashed). **b**, Fraction of risk of an ACE > 178 season attributable to the SST difference between the 2020's SST levels and 1982's SST levels (dashed lines). The histograms show the fraction of attributable risk (FAR) distributions from a 10000-member bootstrapping for the 1982 SST level scenario, the vertical lines indicate the median FARs.

has been argued, that this effect of relative SSTs is the dominant contribution to TC activity (Sobel et al., 2016; Murakami et al., 2018). The temperature difference between the tropical Atlantic and other tropical basins has a strong impact on atmospheric circulation. However, our sensitivity analysis suggests, that SSTs over the MDR contain relevant information and that our approach to include SSTs in the emulator as an addition to the sequences of weather patterns is suited to simulate intense TCs.

The potential maximum intensity a TC can reach depends on the temperature difference between the ocean surface and the tropopause layer and it is plausible, that increasing SSTs have an amplifying effect on strong TCs (Emanuel, 1987). However, it has been argued that over the satellite era, tropopause layer cooling might have dominated over the role of SSTs (Emanuel et al., 2013), a hypothesis we cannot exclude based on our analysis.

Ultimately, the integration of SSTs in our model relies on assumptions that are physically motivated and that lead to a better

representation of TC activity over the period 1982-2020 than other assumptions. Since the early 1980's, an increase in global average surface air temperature of more than $0.5\,K$ has occured and we would argue that over this period SSTs in the region serve as a useful proxy for thermodynamic changes in the climate system.

There is increasing consensus in the scientific literature that the number of tropical cyclones might not or only moderately increase, while the number of most intense storms would increase substantially (Masson-Delmotte et al., In Press). Our emulator results indicate that increasing SSTs could be a potential driver for such an intensification, also allowing for potential avenues to link those changes more directly to anthropogenic climate change.

In this first application of the emulator we have focused on ocean warming. Applying the emulator to future climate projections from state of the art earth system models might, however, also help to estimate the dynamic forcing on TC activity resulting from atmospheric circulation changes. While most climate models have a poor representation of TCs, their projections of atmospheric circulation changes contain valuable information that could be meaningfully analyzed using this TC emulator.

By separating out the thermodynamic and dynamic forcings for observed ACE, our approach allows us to link the observed trend in seasonal cyclone activity and extreme season probability to warming SSTs. Our findings indicate that warming SSTs over the tropical Atlantic might have already contributed significantly to more extreme tropical cyclone seasons, and thereby to the fatalities, destruction and trillion dollar losses that these cyclones have caused over this the last four decades (MunichRe, 2021). Given the projected increases of SSTs with increasing warming, our findings suggest that the probability of extreme seasons might further increase. To minimise future risks, stringent emission reductions in line with achieving the goals of the Paris Agreement would be required (Masson-Delmotte et al., In Press).

*Code availability.* All python scripts required to perform the analysis and create the plots is available under https://zenodo.org/record/6223723.

*Author contributions.* P.P. and C.-F.S. conceived the study. P.P. developed the methods with contributions from C.-F.S. and S.N.. P.P. prepared all figures. P.P. wrote the manuscript with contributions from all authors.

*Competing interests.* No competing interests.

*Acknowledgements.* We acknowledge the NOAA's International Best Track Archive for Climate Stewardship (IBTrACS), the fifth generation of ECMWF atmospheric reanalyses (ERA5), the Optimum Interpolation Sea Surface Temperature (or daily OISST) from NOAA and the Hadley Centre Sea Ice and Sea Surface Temperature data set (HadISST). P.P. and C.-F.S. acknowledge support by the German Federal Ministry of Education and Research (01LN1711A). S.N. was supported by the LAMACLIMA project, receiving funding from the German Federal Ministry of Education and Research (BMBF) and the German Aerospace Center (DLR) as part of AXIS, an ERANET initiated by JPI Climate (grant no. 01LS1905A), with co-funding from the European Union (grant no. 776608).

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
