# Peer review of "Extreme Atlantic hurricane seasons made twice as likely by ocean warming"

_Weather and Climate Dynamics, 2021_

## Author Comment (AC2)

Response to Anonymous Referee #2

This manuscript assesses whether extreme hurricane seasons can be attributed to ocean warming and changes to circulation patterns. The authors develop a novel statistical model that utilizes daily weather patterns and average SSTs and find that increases in Atlantic SSTs have led to a higher likelihood that the North Atlantic will have extremely active hurricane seasons (or more extreme hurricanes). This paper is worthy of publication after some major changes to the delivery. There are a few areas worth mentioning.

1. I think the manuscript needs to be placed in the context of the literature. The authors cite a few papers related to the topic but there are many others. See my comments below.

*We thank the reviewer for the concrete suggestions and will incorporate them in the revised manuscript.*

2. I think the organization of the supplement needs some work. It included a large amount of information and introduced figures in a nonintuitive order. See my comments below.

*We agree that the supplementary information could be organized and structured more meaningfully.*

Minor Comments:

- Line 49: If the data are available, I'd recommend including 2019 and 2020 in your dataset.

*We will do so in the revised manuscript.*

- Line 53: Are the grid cells used to average those that are directly around (nearest0-neighbor) or is there a farther extent used for the average? Please clarify this in the text. And what is the temporal resolution of the data you used to average back out to daily? Did it start as daily data?

We would clarify this as follows in the revised manuscript:
"In order to remove the direct influence of TCs in the reanalysis data we replace an area of $9°x9°$ around the center of the storm with an average of the 16 grid-cells that lie outside of this 3x3 grid-cell square around the storm center."
*The original data is sampled on a 6-hourly frequency. We will clarify this in the revised manuscript.*

- Why are the tropical north Atlantic tropical storms being taken from a slightly larger area than the SST data? It would make more sense to have the same region unless there is a reasonable explanation for why it should remain different. It should be mentioned in the text.

*We agree that this inconsistency should be explained in the text. While we want to study all tropical storms in the Atlantic basin, most of the storms form over a smaller region, the so called main development region (MDR). This region is commonly used in the literature and we would like to stick to this region as it reflects quite well the relevance of SSTs on TCs in the Atlantic.*
*We initially used a region for storm selection (fig. S3) to filter out extratropical cyclones. In the revised manuscript we will not use this region anymore and instead use the storm classification from IBTrACS to filter out extratropical cyclones.*

- Figure S3 and S7 could be greatly improved. There are no axes labels, north arrows, scale bars. I think the authors should spend some time here.

*We will improve both figures including grid lines and coordinate labels on the axes.*

- Line 55: check acronym for ITrACS (which should also be spelled out completely for first use).

*Well noted.*

- What is your definition of an event? Is it all storms over 34 kts? 64 kts? Please add to text.

*In the initial version, we included all storms for the ACE calculation. In the revised version we will only include storms above 34 kts and will clarify this in the text.*

- How many storms does your analysis include? Please add near line 64.

*In the revised manuscript we will include the years 2019 and 2020. With those years, the analysis will include 806 storms. We will include this information in the revised manuscript.*

- Where are you getting your equation for ACE? Please cite.

*We will cite Bell et al. 1999 in the revised manuscript.*

- You use many acronyms without introducing them first – WMO, NOAA, IBTrACS. Please look through and introduce the first and then use them again after that. Just helps to make the reading a bit easier.

*Thank you for pointing that out. We will introduce all acronyms in the revised manuscript.*

- Your supplementary material figure order isn't intuitive. I'm getting S7 in the text before S1.

*We will restructure the supplementary material in a more meaningful way.*

- Figure 1 – I love this! I think you made an excellent figure and a great way to visualize the variables across time. Once this is published, I'll be using this in my hurricane class. Nicely done!

*Thanks a lot! We would be happy if you use it.*

- I'm not sure I understand the purple arrow placement on Figure 2. Can you add something to the text that describes this figure to guide a reader how to use it.

*The purple arrow implies that one input for the intensity calculation is the intensity of the storm on the day before. This input can not directly be related to SSTs or weather patterns. We will clarify this in the figure caption.*

- Line 116: extent, not extend

*Well noted.*

- Paragraph 112-115: I'm not sure I understand the choices you have made for the duration. At the very least, this section needs to be cited for support about why the duration assumptions are

made. Even better would be a sensitivity to test to understand how sensitive your model output is to different assumptions. This is also applicable to your intensity discussion.

*Please see the comment 6 comments below that starts with: "Disregard my comment above about sensitivity testing..."*

- Line 134: too not to

*Well noted.*

- Lines 112-135 need to be better cited. There are many publications that can be used to show support for these things. Trepanier (2020) in atmosphere utilizes local SSTs in the North Atlantic to estimate the probability of extreme hurricane winds throughout the basin. This is applicable to support your choices here.

*We agree that a citation for this paragraph was missing and thank the reviewer for suggesting this reference. We will include it in the revised manuscript.*

- Line 153: Did your data start with 1979 as previously stated or 1982? Please check this.

*We thank the reviewer for pointing out this inconsistency. When starting the clustering exercise for the weather patterns, we were working on the available ERA5 data at that time which was 1979-2018. Later on we decided to use SSTs from the DOISST data set from NOAA, which only starts in 1982. In a revised manuscript we would consistently use the period 1982-2020 (or even 2021) for each part of the analysis.*

- In Figure 3, what is your neutral year?

*Thanks for pointing out this inconsistency! The neutral year (cyan in fig. 3) is 2009. We will change the figure caption and the text accordingly.*

- It is interesting that ACE has the highest correlation and storm formation has the least. This could be related to the way you defined storm formation (and duration) but it could also be related to the notion that SST more predominantly affects intensity and less the formation of storms. Perhaps worth mentioning.

*We thank the reviewer for sharing this interpretation. In our view, this mainly shows that with weather patterns and SSTs we can adequately estimate the potential for seasonal TC activity which appears to be easier than emulating storm formations. In that sense, the intensification component of the emulator can be seen as control instance that won't allow seasons to become highly active if SSTs are cool even if for some reason the emulator produced a high number of storms. Consequently, one could think that including SSTs into the storm formation component could improve the skill of the emulator, but there is no clear evidence for the effect of SSTs on storm formation numbers and our sensitivity analysis suggests, that including SSTs in the storm formation component does not improve the emulator. In the revised manuscript we will elaborate more on this aspect.*

- Disregard my comment above about sensitivity testing. I see you did this in the supplementary. Kudos and thanks for that. I still think it needs additional citations, though, pretty much throughout this whole thing.

*We agree with the reviewer and will add references for the assumptions underlying the three components of the emulator.*

- The supplement is difficult to follow and dense. As I was finishing this review, I noticed Reviewer 1 commented. I completely second this individual's 2nd main comment. The amount you have in the supplement looks to be enough for another manuscript. It should be easy to follow and directly relate to the text. You may reconsider restructuring it.

*We totally agree with both reviewers and will restructure the supplement.*

- Figure 6 caption description – I think you mean vertical line, not horizontal

*Thank you!*

- Since 2019 and 2020 are not represented in here, as I noted earlier, can your emulator be used to estimate the number of storms, etc., for those two seasons? Can you put the known conditions in and see if it produces a similar ACE, duration, etc. for those that aren't in the data set? Could be an interesting addition to the work.

*We will include those years for the analysis of the revised manuscript.*

- What are you hoping someone does with this emulator? Can you provide a little broader impact context to the discussion?

*We are hoping that this emulator can be used to further investigate the influence of potential changes in atmospheric circulation patterns on hurricane activity. One major challenge in tropical cyclone research is that TCs are not well reproduced in common climate models. Applying this emulator to projections of weather patterns and SSTs from a comprehensive ensemble of climate models could help to better estimate how atmospheric circulation changes resulting from global warming might affect TC activity.*

---

## Author Response (AR1)

**Main changes in the revised manuscript**

For the revision we added a refined analysis of historic SST trends that was in part motivated by a question of the anonymous reviewer 1.

In the initial submission we assume that there has been a linear warming of SSTs in the MDR for the period 1870-2018. However, global mean temperatures did not increase linearly over this long period and especially tropical Atlantic SSTs only started substantially warm after 1980. Little ocean surface warming in the MDR has occurred prior to 1980 (see fig. 5 in the revised manuscript).

In the revised manuscript we therefore focus on the forced SST trend over the period 1982-2020. We estimate this trend from CMIP6 historical simulations over the the period 1982-2014. We then remove this forced trend from the SST observations to get a time series of natural variability over the period 1982-2020. The counterfactual scenarios now consist of this detrended SST data shifted to the values of the CMIP6 trend at the years 1982 and 2020 (see fig. 5 in the revised manuscript).

It turned out, that there has been little forced SST warming in the MDR prior to 1982 (see figure 5 and lines 209-222). In the revised manuscript there are therefore only two counterfactual SST scenarios: one representing 1982's SST levels and one representing 2020's SST levels (and no "pre-industrial scenario). This makes our statements concerning the effect of SST warming since 1982 on TC activity considerably stronger. Changes of statements throughout the results section and in the abstract are a result of this better founded construction of counterfactual SST scenarios which in our view substantially improves the analysis.

**Response to Anonymous Referee #1**

This manuscript explores the question of whether recent extreme hurricane seasons can be attributed to changes in circulation and SST. To explore this question, the authors use a new statistical model to explore weather patterns and find that increases in Atlantic SSTs have led to an increase in the probability of extremely active hurricane seasons. While I think that this is an interesting result worthy of publication, I believe that in its current form the manuscript has two major issues:

1. There are several inconsistencies throughout the manuscript (see examples below), as it pertains to the time periods and regions used for the analysis. Some additional clarity is needed and urge the authors to check the full manuscript for consistency.

We agree with the reviewer and are thankful for them pointing out the above mentioned inconsistencies. Practical choices that made sense during the process of our explorative analysis led to inconsistencies in the previous version of the study and we are more than happy to provide a fully consistent methodology in the revised version of the manuscript. Furthermore, it appears that clearing out the rightfully mentioned inconsistencies does affect the results and conclusions of the paper.

2. In its current form, I find the organization and utility of the supplementary information figures difficult to follow. While there are references to the figures throughout the main manuscript, the

order - and need for 28 additional figures - is not clear and distracts from the main results of the manuscript. I would encourage the authors to include all necessary figures for interpreting the results in the main manuscript and use supplementary information plots to contextualize statements. In addition, all Figures should be able to stand on their own and be interpretable. I find the lack of axis markers and labels limit the effectiveness of some figures.

We agree that the supplementary information of the initial submission could be organized and structured more meaningfully. In a revised version we leave out a few figures in the supplement and improve the remaining ones. We also included one figure about SST trends in the manuscript. We are however hesitant to include more figures in the manuscript. We carefully selected the content that we want to show in the main paper as we don't want to overload the paper with figures that might distract the reader.

Once these major comments, as well as the list of minor comments below, are taken into consideration, I believe the manuscript would be suitable for publication in Weather and Climate Dynamics.

**Additional Comments:**

L49: Why is the 1979-2018 period used? In doing so, the authors are leaving out the 2019 and 2020 seasons, which were destructive. Does the signal become larger? If so, this would be very interesting to include.

We thank the reviewer for pointing out this inconsistency. When starting the clustering exercise for the weather patterns, we were working on the available ERA5 data at that time which was 1979-2018. Later on we decided to use SSTS from the DOISST data set from NOAA, which only starts in 1982. In the revised manuscript we consistently use the period 1982-2020 (or even 2021) for each part of the analysis.

L53: What is the original frequency that is averaged to daily?

The original data is sampled on a 6-hourly frequency. We clarify this in the revised manuscript (L56).

L55: Why is the SST averaging area (20W-90W) different than the area used to classify the weather patterns (10W-90W)? Is the analysis sensitive to this approach?

Our weather patterns are supposed to represent the large scale atmospheric circulation in the region while the region over which we average SSTs is the hurricane main development region (MDR) that is commonly used in the literature. These two regions serve a different purpose for the analysis. We think that it makes sense to stick to the used region for the SST averaging to facilitate comparability with other studies. The region for the weather pattern classification could be altered, but we do not expect this to considerably affect the weather patterns.

**L57: IBTrACKS should be IBTrACS.**

**Well noted.**

L58: How is this region in Figure S3 selected? Why not all of the Atlantic? Is the results sensitive to this selection?

We initially used this region to filter out extratropical cyclones. We now checked again and found out, that the type of storm is documented in IBTrACS. In the revised manuscript we use this storm classification from IBTrACS and won't need this region any more.

L60: When calculating ACE, do the authors follow Bell et. al, 2000 and only include storms while they are tropical storm strength or great? That should be clarified since the NOAA classification for above normal and extremely active seasons. Are the results sensitive to this methodology?

We thank the reviewer for pointing out this inconsistency! In the initial manuscript we calculated ACE including all storms, also those which are weaker than tropical storm strength. In the revised manuscript we change that to be consistent with Bell et al. 2000. As the seasonal ACE is calculated in the same way for observations and emulations we do not see major effects on the results.

L93: Missing "and" before iii)

Well noted.

Figure 2: States SST areas is 85W-20W, 10N-20N which is different than the in-text description.

Apologies for this typo! The used region is 90W-20W, 10N-20N. We updated the figure accordingly.

**L150: How is major hurricanes defined?**

**The definition can be found in L74 (previously L62):**

"TCs are classified according to the Saffir-Simpson hurricane wind scale according to which TCs with sustained winds of more than 64 knots are named hurricanes and TCs with sustained winds above 96 knots are major hurricanes."

Figure S1-3, S5 and S7: Should include axis (lat-lon) markers.

**Is included in the revised manuscript.**

L152: Why is 1982-2018 used here, when it was stated earlier that 1979-2018 is used for classification of weather patterns?

We have answered this question in our reply to the comment about L49 above.

L195: Why is 1900 used, if the long-term SST trend calculation starts with 1850?

This is a good question. Looking at figure S9, we would argue that the SST levels would be quite similar for 1900 and 1850. This however points to more important flaw in our way of constructing counterfactual SST scenarios: applying a linear trend from 1850 to 2020 is too simplistic and does not reflect our understanding of global warming. See our explanation at the beginning of this document.

**L230-234: What about all years?**

The probability of finding 225 ACE irrespective for the weather patterns and SST variations of all observed seasons also increases by roughly 30% as a result of the SST warming since the 1980s (see dashed cyan line in fig. 6b). We add the respective numbers for 2020 in the revised manuscript.

L240-241: What about dust impacts on these weather patterns and potential trends?

This is a good question and we would be interested in investigating this further. Changes in atmospheric circulation patterns are, however, beyond the scope of this paper.

**Response to Anonymous Referee #2**

This manuscript assesses whether extreme hurricane seasons can be attributed to ocean warming and changes to circulation patterns. The authors develop a novel statistical model that utilizes daily weather patterns and average SSTs and find that increases in Atlantic SSTs have led to a higher likelihood that the North Atlantic will have extremely active hurricane seasons (or more extreme hurricanes). This paper is worthy of publication after some major changes to the delivery. There are a few areas worth mentioning.

3. I think the manuscript needs to be placed in the context of the literature. The authors cite a few papers related to the topic but there are many others. See my comments below.

We thank the reviewer for the concrete suggestions. We have added the suggested references and a few additional ones.

4. I think the organization of the supplement needs some work. It included a large amount of information and introduced figures in a nonintuitive order. See my comments below.

We agree that the supplementary information could be organized and structured more meaningfully. We have fundamentally restructured the supplementary information. We only show relevant figures and arranged them in a more intuitive way.

**Minor Comments:**

• Line 49: If the data are available, I'd recommend including 2019 and 2020 in your dataset. *We added the years 2019 and 2020 in the revised manuscript.*

• Line 53: Are the grid cells used to average those that are directly around (nearest0-neighbor) or is there a farther extent used for the average? Please clarify this in the text. And what is the temporal resolution of the data you used to average back out to daily? Did it start as daily data?

We clarified this as follows in the revised manuscript (L54-56): "In order to remove the direct influence of TCs in the reanalysis data we replace the 3x3 grid-cell square area encompassing the center of the storm with the average of its surrounding 16 grid cells." *The original data is sampled on a 6-hourly frequency. We clarified this in the revised manuscript.*

• Why are the tropical north Atlantic tropical storms being taken from a slightly larger area than the SST data? It would make more sense to have the same region unless there is a reasonable explanation for why it should remain different. It should be mentioned in the text.

We agree that this inconsistency should be explained in the text. While we want to study all tropical storms in the Atlantic basin, most of the storms form over a smaller region, the so called main development region (MDR). This region is commonly used in the literature and we would like to stick to this region as it reflects quite well the relevance of SSTs on TCs in the Atlantic. We initially used a region for storm selection (fig. S3) to filter out extratropical cyclones. In the revised manuscript do not use this region anymore and instead use the storm classification from IBTrACS to filter out extratropical cyclones.

• Figure S3 and S7 could be greatly improved. There are no axes labels, north arrows, scale bars. I think the authors should spend some time here.

*We improved these figures including grid lines and coordinate labels on the axes.*

• Line 55: check acronym for ITrACS (which should also be spelled out completely for first use). *Well noted.*

• What is your definition of an event? Is it all storms over 34 kts? 64 kts? Please add to text. *In the initial version, we included all storms for the ACE calculation. In the revised version we only include storms above 34 kts and clarified this in the text (L72-73).*

• How many storms does your analysis include? Please add near line 64. In the revised manuscript we will include the years 2019 and 2020. With those years, the analysis will include 454 storms. We included this information in the revised manuscript (L69).

• Where are you getting your equation for ACE? Please cite. *We cite Bell et al. 1999 in the revised manuscript (L69).*

• You use many acronyms without introducing them first – WMO, NOAA, IBTrACS. Please look through and introduce the first and then use them again after that. Just helps to make the reading a bit easier.

Thank you for pointing that out. We introduced all acronyms in the revised manuscript.

• Your supplementary material figure order isn't intuitive. I'm getting S7 in the text before S1. *We restructured the supplementary material in a more meaningful way.*

• Figure 1 – I love this! I think you made an excellent figure and a great way to visualize the variables across time. Once this is published, I'll be using this in my hurricane class. Nicely done!

Thanks a lot! We would be happy if you use it.

• I'm not sure I understand the purple arrow placement on Figure 2. Can you add something to the text that describes this figure to guide a reader how to use it.

The purple arrow implies that one input for the intensity calculation is the intensity of the storm on the day before. This input can not directly be related to SSTs or weather patterns. We clarified this in the figure caption.

• Line 116: extent, not extend *Well noted*.

• Paragraph 112-115: I'm not sure I understand the choices you have made for the duration. At the very least, this section needs to be cited for support about why the duration assumptions are made. Even better would be a sensitivity to test to understand how sensitive your model output is to different assumptions. This is also applicable to your intensity discussion.

*Please see the comment 6 comments below that starts with: "Disregard my comment above about sensitivity testing..."*

• Line 134: too not to *Well noted*.

• Lines 112-135 need to be better cited. There are many publications that can be used to show support for these things. Trepanier (2020) in atmosphere utilizes local SSTs in the North Atlantic to estimate the probability of extreme hurricane winds throughout the basin. This is applicable to support your choices here.

We agree that a citation for this paragraph was missing and thank the reviewer for suggesting this reference. We include it in the revised manuscript alongside with Bhatia et al. 2018 (L136).

• Line 153: Did your data start with 1979 as previously stated or 1982? Please check this. We thank the reviewer for pointing out this inconsistency. When starting the clustering exercise for the weather patterns, we were working on the available ERA5 data at that time which was 1979-2018. Later on we decided to use SSTs from the DOISST data set from NOAA, which only starts in 1982. In the revised manuscript we consistently use the period 1982-2020 (or even 2021) for each part of the analysis.

• In Figure 3, what is your neutral year?

Thanks for pointing out this inconsistency! The neutral year (cyan in fig. 3) is 2009. We changed the figure caption and the text accordingly.

• It is interesting that ACE has the highest correlation and storm formation has the least. This could be related to the way you defined storm formation (and duration) but it could also be related to the notion that SST more predominantly affects intensity and less the formation of storms. Perhaps worth mentioning.

We thank the reviewer for sharing this interpretation. In our view, this mainly shows that with weather patterns and SSTs we can adequately estimate the potential for seasonal TC activity which appears to be easier than emulating storm formations. In that sense, the intensification component of the emulator

can be seen as control instance that won't allow seasons to become highly active if SSTs are cool even if for some reason the emulator produced a high number of storms. Consequently, one could think that including SSTs into the storm formation component could improve the skill of the emulator, but there is no clear evidence for the effect of SSTs on storm formation numbers and our sensitivity analysis suggests, that including SSTs in the storm formation component does not improve the emulator. In the revised manuscript we elaborated a bit on this aspect (L179-183).

• Disregard my comment above about sensitivity testing. I see you did this in the supplementary. Kudos and thanks for that. I still think it needs additional citations, though, pretty much throughout this whole thing.

We agree with the reviewer and added references for the assumptions underlying the three components of the emulator.

• The supplement is difficult to follow and dense. As I was finishing this review, I noticed Reviewer 1 commented. I completely second this individual's 2nd main comment. The amount you have in the supplement looks to be enough for another manuscript. It should be easy to follow and directly relate to the text. You may reconsider restructuring it.

We totally agree with both reviewers and restructured the supplement.

• Figure 6 caption description – I think you mean vertical line, not horizontal *Thank you!*

• Since 2019 and 2020 are not represented in here, as I noted earlier, can your emulator be used to estimate the number of storms, etc., for those two seasons? Can you put the known conditions in and see if it produces a similar ACE, duration, etc. for those that aren't in the data set? Could be an interesting addition to the work.

We included those years for the analysis of the revised manuscript.

• What are you hoping someone does with this emulator? Can you provide a little broader impact context to the discussion?

We are hoping that this emulator can be used to further investigate the influence of potential changes in atmospheric circulation patterns on hurricane activity. One major challenge in tropical cyclone research is that TCs are not well reproduced in common climate models. Applying this emulator to projections of weather patterns and SSTs from a comprehensive ensemble of climate models could help to better estimate how atmospheric circulation changes resulting from global warming might affect TC activity. We added a paragraph on this in the discussion (L276-279).

---

## Author Response (AR2)

Again, this manuscript explores the question of whether recent extreme hurricane seasons can be attributed to changes in circulations and SST. To explore this question, the authors use a new statistical model to explore weather patterns. They find that increases in Atlantic SSTs have led to an increase in the probability of extremely active hurricane seasons. It is clear from the revised manuscript that the authors have improved presentations since the last submission. However, before the manuscript is suitable for publication in Weather and Climate Dynamics there are several minor comments that still need to be addressed:

We thank the reviewer for reviewing a second time. We appreciate the useful comments, questions and suggestion that in our view strongly improved the manuscript!

L7: From the results section, I am a little unclear of where the number 50% comes from. Figure 6e? This percentages should be directly calculated in the results section (i.e., L228 should provide the value for the 2020 counterfactual) to be mentioned in the abstract. Same is true for the "doubled the probability of extremely active tropical cyclone seasons" statement in the previous sentence.

We agree, that this number is a bit misleading, especially as the next number is given as a factor. As we are more concerned about extremely active seasons we removed this statement from the abstract. We nevertheless reformulated parts of the result section to avoid confusion with factors and percentage points:

'Consequently, more than one third of the seasons that are simulated to be above normal seasons in the 2020 scenario are below normal seasons in the 1982 scenario (an above normal seasonal activity being defined as > 126.1 ACE (CPC, 2021)). Similarly, the number of simulations that are classified as extremely active (with ACE > 159.6) doubles from 11% in the 1982 scenario to 22% in the 2020 scenario (see fig. 6g).'

The part in the abstract is now:

'For the year 2020, our results suggest that such an exceptionally intense season might have been made twice as likely.'

**L56: ERA5 has a 30km horizontal grid resolution not 1 deg., correct? Please clarify.**

That is correct. We used the option of the copernicus climate data store (https://cds.climate.copernicus.eu) to download the data on a 1°x1° grid. We agree, that this pre-processing step should be transparent in the manuscript. For better readability, we explain the procedure as follows:

'For the following pre-processing we transform the data from the original  $0.28^{\circ} \times 0.28^{\circ}$  to a  $1^{\circ} \times 1^{\circ}$  grid. In order to remove the direct influence of TCs in the reanalysis data we replace the 3x3 grid-cell square area encompassing the center of the storm with the average of its surrounding 16 grid cells. Finally, we transform the data from a  $1^{\circ} \times 1^{\circ}$  grid to a  $2.5^{\circ} \times 2.5^{\circ}$  grid and average 6-hourly data to daily data.'

*L60:* Note, that the MDR displayed in Figure S6 is 10N-20N, not 10N-30N as stated here in the text. This should be consistent. Which is used for the analysis?

We thank the reviewer for noting this inconsistency! The definition of the MDR that we use in the analysis is 10N-20N. We corrected this in the manuscript.

**L87: Why 1982-2011? This is different than the period mentioned in Section 2.1.**

For the standardization of the variables we want to use a period that is at least 30 years long. As we are planning to use the emulator for an analysis based on CMIP6 simulations we

thought it would be convenient if the period that we use for standardization is fully covered by the historic simulations which only reach 2014. In that sense the period chosen here is an arbitrary 30 year period. We do not expect that choosing a different period for the standardization would significantly alter the results.

**L95: Should this reference figure S7?**

**Indeed, we wanted to reference figure S7.**

L170: The authors could consider including a large-scale predictor of these storm precursors. Hsieh et al. 2020 (https://doi.org/10.1007/s00382-020-05446-5) showed that vertical velocity is a good predictor for the number of convective cluster and that the ratio of the local Rhines scale and the Rossby deformation radius is a good predictor of the number of these convective clusters that become weak rotating systems. This might be a good direction for future work.

We thank the reviewer for this relevant reference. This might indeed be helpful for future work. In the current design of the emulator we do not resolve the location of storms which might be one of the major caveats. This makes the inclusion of indicators such as vertical velocities challenging. Now, that we have developed this simple emulator, it would however be interesting to think of ways that would allow to include the location of storms or more indicators that would help to better represent storm formations.

**L206: Any comment on why the HadISST seems to have warmed less (Figure 5a should 0.23 K per decade) over this time period than the DOISST?**

We agree that this is worth mentioning here, even though we already commented on this in line 220 of the previous manuscript. We added in line 206: 'This trend is slightly weaker in the HadISST dataset (figure 5a) for which also the global trend in SSTs over the period 1982-2020 is weaker than in other SST datasets (Yang et al., 2021).'

Figure 5c: The link colors for the 1982 and 2020 counterfactuals should be the same for Figures 5 and 6. Also, the legend should state 1982 and 2020, not 1980's and 2020's to be consistent.

We thank the reviewer for pointing out these inconsistencies. We updated the figure accordingly.

**L220: I would again specify that this is true for the MDR: "...pre-industrial levels for the MDR."**

In the revised manuscript, this is clarified as suggested by the reviewer.

Figure 6a: This panel is never referred to in the text and the caption states that the observed value is displayed in black, which it is not. Also, there is only two counterfactual scenarios not three as stated in the caption.

We updated the figure caption according to what is shown in the panel. We were also thinking about dropping the panel as the counterfactual SST scenarios are indeed already shown in figure 5c. We concluded that it is still helpful to have these SST time series in figure 6 to allow for an easier interpretation of the counterfactual simulations in the following panels.

**Figure 6b: Why are some purple boxes a lighter shade?**

These are years where the seasonal SST averages in the counterfactual scenario lie outside of the range of observed seasonal SST averages. We added a clarifying sentence in the figure caption: 'For years, where the seasonal SST averages in the counterfactual scenario are outside of the range of observed seasonal SST averages, the simualtions are shown in lighter shading.'

*Figure 7: 1983 is not mentioned in the caption. Also, this figure should use different coloring than Figures 5 and 6, so as to not confuse the reader.*

We agree with the reviewer and changed the caption and the colors accordingly.

L240: Consider adding a vertical lines to Figure 7a for "above normal season" and "extreme active seasons."

Thanks for the suggestion. We added these lines.

**L244-246: What about for the observed 178 ACE?**

The probability of finding 178 ACE with 1982's SST levels is 21\% according to our simulations. We clarified this in the revised manuscript.

**L250-251: Why is it higher for all years?**

If favorable weather conditions lead to many storm formations in a season there is a considerable likelihood of reaching 178 ACE irrespective of the SST levels. For instance, for 2005 half of the simulations with 1982's SST levels are extremely active seasons and roughly 40% of the simulations reach 178 ACE. While in our simulations warmer SSTs will further intensify these extremely active seasons, this will not affect the increased risk of getting more than 178 ACE. Therefore, the increased risk of finding seasons with more than 178 ACE due to SST warming is more pronounced for years with lower TC activity. (In the extreme case, where only a few simulations with 2020's SST levels reach 178 ACE and none of the simulations with 1982's SST levels, the calculation of the increased risk would require a division by zero.)

We included a comment in the revised manuscript:

'The increase in likelihood of finding 178 ACE is higher for seasons with weather conditions that are hampering TC formation and development.'

L270: What is this 0.5 K of global warming referring to? Global average surface temperature? Figure 5b. shows that the MDR has warmed by over 1.1 K over this period, right? This sentence should be clarified.

We agree, that this sentence could be misleading. In the revised manuscript we clarified, that we are writing about an increase in global average surface air temperature.